# Improving Drying Characteristics and Physicochemical Quality of *Angelica sinensis* by Novel Tray Rotation Microwave Vacuum Drying

**DOI:** 10.3390/foods12061202

**Published:** 2023-03-12

**Authors:** Zepeng Zang, Xiaopeng Huang, Cuncai He, Qian Zhang, Chunhui Jiang, Fangxin Wan

**Affiliations:** College of Mechanical and Electronical Engineering, Gansu Agricultural University, Lanzhou 730070, China

**Keywords:** microwave vacuum drying, *Angelica sinensis*, drying characteristics, physicochemical quality, microstructure

## Abstract

In order to improve the shortcomings of uneven heating of traditional microwave drying and to maximally maintain food quality after harvest, a rotary microwave vacuum drying equipment was fabricated and used for drying experiments on *Angelica sinensis* to explore the effects of drying temperature, slice thickness, and vacuum degree on drying characteristics, physicochemical quality, and microstructure of dried *Angelica sinensis* products. The results showed that microwave vacuum drying can significantly shorten the drying time and improved the drying efficiency. Six different mathematical models were investigated and the Midilli model was the best-fitted model for all samples (R^2^ = 0.99903, Pearson’s r = 0.99952), and drying methods had various effects on different indexes and were confirmed by Pearson’s correlation analysis and principal component analysis. The optimal process parameters for microwave vacuum drying of *Angelica sinensis* were determined by entropy weight-coefficient of variation method as 45 °C, 4 mm, −0.70 kPa. Under this condition, well preserved of ferulic acid, senkyunolide I, senkyunolide H, ligustilide, total phenols and antioxidant activity, bright color (L* = 77.97 ± 1.89, ΔE = 6.77 ± 2.01), complete internal organizational structure and more regular cell arrangement were obtained in the samples. This study will provide a theoretical reference for the excavation of the potential value and the development of industrial processing of *Angelica sinensis*.

## 1. Introduction

*Angelica sinensis* (Oliv.) Diels (Family Umbelliferae) is a perennial traditional Chinese medicine distributed mainly in cool-moist regions of western China [1]. Its chemical constituents mainly include phthalide lactones, organic acids, polysaccharides, amino acids, and nucleosides, etc. Recently, *Angelica sinensis* was shown to have the function of lowering atherosclerosis and thrombosis, anticancer, antioxidant, and cardioprotective and protecting nerve cells, and it has significant effects in protecting chondrocytes and treating Alzheimer’s disease [2,3]. Freshly harvested *Angelica sinensis* contain a high quantity of free and bound water, approx. 60~70% in wet basis (w.b.). High moisture content may enhance enzymatic and nonenzymatic reactions easily and result in rapid deterioration of product physicochemical quality and reduction in nutritional and commercial value [4,5].

Drying is one of the most effective methods to prolong the shelf life of food by reducing moisture content and water activity to prevent the reproduction of microorganisms, deterioration reaction, and enzyme activity [6,7,8]. However, drying is a complex process of heat and mass transfer, which may lead to the loss of nutrients and unnecessary structural changes in the process of dehydration, thereby affecting the color, flavor, and nutritional value [4,9]. Therefore, appropriate drying methods play a crucial role in enhancing the medicinal value in *Angelica sinensis*. At present, the research on *Angelica sinensis* drying process is still at its nascent stage. Sun drying and shade drying are the traditional drying method in *Angelica sinensis*, which are low cost and widely applicable but have shortcomings such as being time-consuming and the poor quality of dried products, thus resulting in a reduction in nutrition value [10]. In recent years, the demand for high-quality food has led to the development of a number of new drying methods, such as hot air drying, far-infrared drying, and freeze drying; each has its unique advantages and limitations. Due to fast heating and simplicity, drying with hot air has become the most widely used method. However, this drying method has some disadvantages, such as causing the degradation of nutritional compounds and the Maillard reaction [11]. Far-infrared radiation drying is considered a promising method for food and pharmaceutical product dehydration, and due to its ability to significantly shorten the drying time and retain nutrients [12]. However, it will cause uneven heating of materials and poor rehydration properties. Freeze drying has many benefits, including preventing wrinkling and shrinkage, and maintaining bioavailability and flavor of the dried products [13]. Nevertheless, it is an expensive and high energy consumption method that limits its applications. Radha et al. [14] studied the effects of different drying conditions on the antioxidant activity and color of apple slices, and found that ultrasonic vacuum drying accelerated the drying rate and improved the physicochemical quality of the samples. Ahmad et al. [15] evaluated the effect of different drying treatments on the physicochemical properties and antioxidant activity of kiwifruit, and the results showed that the highest content of bioactive compounds was found in the samples after mixed drying. The above results indicate that a reasonable drying technology and conditions can inactivate microorganisms, and to maximally retain sensory and physicochemical characteristics, taste, color, and flavor of the dried product, and play a vital role in improving the edible value and medicinal value of food [16].

Microwave drying is volumetric heating (from inside out) that takes place as a result of electromagnetic waves penetrating directly into the material causing molecular friction, and this molecular friction generates thermal energy that is then utilized to remove moisture [17,18,19]. Both moisture and thermal gradients move in a similar direction. Compared with conventional heating, microwave drying was a more efficient and green drying method [20,21]. Huang et al. [22] reported that microwave vacuum drying of Stevia could improve the stevioside and rebaudioside A contents of dried products. Sun et al. [23] found that microwave drying was a potential method to enhance the physicochemical quality and utilization of cajanus cajan. Parveez et al. [24] showed that the anthocyanin, antioxidant capacity, ascorbic acid, and VC content had high values for blueberries after microwave drying. However, non-uniform heating and low porosity are also the main disadvantages of microwave drying, this causes surface hardening and local overheating that damages the nutritional value [25,26]. To overcome this problem could be by coupling additional energy sources. The microwave vacuum drying is a novel technology for drying food and pharmaceuticals. It combines microwave and vacuum technology [27], which can reduce the heating non-uniformity, and vacuum drying allows water to evaporate at a lower temperature than atmospheric pressure, while effectively preventing thermal and oxidative damage to bioactive compounds at subatmospheric pressures, and avoiding thermal-induced degradation of nutrients [28,29]. Accordingly, microwave vacuum drying attracts more and more attention because its high drying speed, contactless heating, less energy consumption, safe, efficient space use, and high retention of heat-sensitive nutrients [30,31]. Up to now, no literature is available about the effects of microwave vacuum drying on the drying characteristics, physicochemical qualities, and microstructure of *Angelica sinensis*.

Therefore, the present study aimed to: (1) evaluate the effect of microwave vacuum drying on moisture migration and drying kinetics in *Angelica sinensis*; (2) determine the quality attributes including color, total phenols, total phenolics, polysaccharides, antioxidant capacity, and six natural active substances content of dried products; (3) establish a more coordinated, compatible and scientific new quality evaluation method (entropy weight-coefficient of variation method); (4) study the overall effects of physicochemical qualities after drying by principal component analysis (PCA) and Pearson’s correlation analysis. Such studies would contribute to optimizing a suitable high-efficiency and uniform heating drying technology to obtain high-quality products, and provide a certain theoretical and scientific basis for the research and application of *Angelica sinensis* drying technology.

## 2. Materials and Methods

### 2.1. Experimental Materials

Materials: Fresh *Angelica sinensis* in this experiment was purchased from Dingxi City (Gansu, China), and the variety was “Mingui No. 1”. A voucher specimen (M1: 20190725GSWYMG1) was deposited in the herbarium of the College of Pharmacy, Gansu University of Chinese Medicine, Lanzhou, China. The initial moisture content of *Angelica sinensis* was measured and determined as (62.47 ± 0.5)% according to the method of Bassey et al. [12]. The samples with a uniform length and complete shape were selected, cleaned, weighed (120 ± 1) g and placed in tray rotation microwave vacuum drying equipment, and recorded every 8 min using the automatic weighing system until the moisture content of the sample was reduced to 10%. Using traditional microwave drying as a control. To ensure the experimental data were accurate, each experiment was repeated three times.

### 2.2. Experimental Reagents

Chlorogenic acid, ferulic acid, ligustilide, senkyunolide H, senkyunolide I, 3-butenylphthalide are all standards. Folin–Ciocalteu’s reagent, 1,1-diphenyl−2-picrylhydrazyl (DPPH), catechin, ascorbic acid, cresol and acetonitrile (HPLC), methanol, and ethanol (analytically pure) were purchased from Chengdu Reflex Biotechnology Co., Ltd. (Sichuan, China).

### 2.3. Experimental Equipments

Tray rotation microwave vacuum dryer: to overcome the problems of traditional microwave drying such as poor drying uniformity, poor controllability and small loading capacity, a novel tray rotation microwave vacuum drying equipment was developed (Figure 1). It mainly consists of drying chamber, control system, transmission system, microwave heating system, and vacuum system. Specify the power of microwave generator is 3 kW, the vacuum degree adjustment range is −60 kPa~−80 kPa, and the rotational speed is adjustable. The external dimension of the drying equipment is 1190 mm × 920 mm × 1580 mm, and the microwave generator is located at the top of the drying chamber, and there are six rotating brackets in the chamber with a rotating diameter of 500 mm. Under the action of the driving device, the material rotates uniformly with the rotating bracket around the spindle and passes through the top microwave source in turn, so as to improve the drying uniformity and quality of the material under short-term and intermittent action.

CR-410 colorimeter, Konica Minolta, Japan; JM-A3003 electronic balance, Henan Yuyao weighing calibration Instrument and Equipment Co., Ltd.; TS-200B incubation shaker, Shanghai Jinwen Instrument and Equipment Co., Ltd.; TGL 20M high-speed centrifuge, Hunan Maijiasen Instrument and Equipment Co., Ltd.; T2600S ultraviolet spectrophotometer, Qingdao Jingcheng Instrument Co., Ltd.

### 2.4. Drying Kinetic Parameters

#### 2.4.1. Moisture Content

The moisture content of the material is calculated according to Equation (1) [32]:(1)$$MR=\frac{M_{t}-M_{e}}{M_{0}-M_{E}}$$
where *MR* represents the moisture ratio of the drying process of the sample; *M_t_* represents the dry base moisture content of the material at any moment *t*, g/g; *M_e_* represents the moisture content of the material at equilibrium, g/g; *M*_0_ represents the initial dry base moisture content of the material, g/g.

#### 2.4.2. Dry Basis Moisture Content

The dry basis moisture content was calculated according to Equation (2) [33]:(2)$$M_{t}=\frac{W_{t}-W_{d}}{W_{d}}$$
where *W_t_* represents the mass of the sample at moment *t*, g; *W_d_* represents the mass of dried, g.

#### 2.4.3. Drying Rate

The drying rate of the dried samples was calculated according to Equation (3) [34]:(3)$$DR=\frac{M_{t}-M_{t-∆t}}{∆t}$$
where *DR* represents the drying rate of the sample during drying, g/(g.min); *M_t_*_1_ and *M_t_*_2_ represent the dry basis moisture content of sample at moments *t*_1_ and *t*_2_, respectively, g/g.

#### 2.4.4. Drying Kinetics

The drying kinetics were studied using six thin-layer models (Table 1), namely, Newton, Midilli, Weibull, Handerson, and Pabis, Logarithmic and Two-term exponential models.

### 2.5. Color

The total color change (ΔE) is one of the most direct indicators to measure the apparent quality of the material [35]. Color variation of *Angelica sinensis* slices was measured by Chao et al. [36] (CR-10 colorimeter, Konica Minolta Ltd., Japan):(4)$$∆E=\sqrt{{\left(L^{*}-L_{0}\right)}^{2}+{\left(a^{*}-a_{0}\right)}^{2}+{\left(b^{*}-b_{0}\right)}^{2}}$$
where *L** represents light/darkness, *a** represents red/greenness, *b** represents is yellow/blueness; *L*_0_, *a*_0_, *b*_0_ represents the initial light/darkness, red/greenness, and yellow/blueness of samples.

### 2.6. Effect of Microwave Vacuum Drying on the Phytochemical Quality

#### 2.6.1. Preparation of Sample Extracts

The dried *Angelica sinensis* was crushed and passed through the No.5 sieve. A 0.5 g sample was accurately weighed and placed in a conical flask containing 20 mL 70% ethanol. The supernatant was centrifuged for 15 min at 4 °C and 6000 r/min (centrifugal radius was 3.5 cm) after shaking for 48 h in a constant temperature shaker whose parameter is 180 r/min under dark conditions.

#### 2.6.2. Determination of Natural Active Ingredients Contents

The dry mass content of natural active ingredients under different pretreatment conditions is carried out with HPLC using the procedure described by Jiang et al. [37] with slight modifications. The chromatographic conditions were: Agilent Eclipse XDB-C_18_ (250 mm × 4.6 mm, 5 μm); mobile phase: acetonitrile (A)-1% acetic acid (B), gradient elution (0~4 min, 85%~85% B; 4~8 min, 60%~35% B; 8~10 min, 35%~15% B; 10~12 min, 15%~80% B; 12~16 min, 80%~85% B); column temperature 25 °C; flow rate 1.0 mL/min; detection wavelength 280 nm; injection volume 10 μL.

#### 2.6.3. Determination of Total Phenols Contents (TPC)

TPC was determined by the Folin–Ciocalteu reagent method [37]. 0.45 mL extract was taken, 2.0 mL 10% Folin–Ciocalteu solution and 1.0 mL 7.5% Na_2_CO_3_ solution were added sequentially. The mixture was shaken for 5 min, and placed in a water bath at 37 °C for 1 h under dark conditions. The absorbance value was measured by spectrophotometer at 765 nm using acidified methanol as a blank control. The total phenolic content was expressed as gallic acid equivalent (GAE), mg/100 g of dry mass. All experiments were performed in triplicate.

#### 2.6.4. Determination of Total Flavonoids Contents (TFC)

TFC was determined by the NaNO_2_-AlCl_3_-NaOH method [38]. A 2.5 mL extract was taken, 2.0 mL distilled water and 0.3 mL 5% NaNO_2_ were added, shaken for 5 min, 0.3 mL 10% AlCl_3_ was added, mixed and shaken for 2 min, and finally 2.0 mL 1 mol/L NaOH was added to make a full reaction. The absorbance value was measured by spectrophotometer at 510 nm using NaNO_2_-AlCl_3_-NaOH solution as a blank control. The total flavonoid content was expressed as rutin equivalent (RE), mg/100 g of dry mass. All experiments were performed in triplicate.

#### 2.6.5. Determination of Polysaccharides Contents

Polysaccharide was determined by the sulfuric acid-phenol method [39]. A 0.005 mL extract was taken, 1.0 mL 9% phenol solution was added, fully mixed, added 3.0 mL concentrated H_2_SO_4_, shaken for 5 min, room temperature reaction for 30 min. The absorbance value was measured by spectrophotometer at 485 nm and without a sample solution as a blank control. The polysaccharide content was expressed as sucrose equivalent (mg/100 g) of dry mass. All experiments were performed in triplicate.

#### 2.6.6. Determination of Antioxidant Activity

Antioxidant activity was determined by DPPH method [39]. An appropriate amount of extract was taken and added to 3.0 mL mol/L DPPH methanol solution, and shaken at room temperature for 30 min under dark conditions. Then, the absorbance A was measured at 515 nm. According to the above operation, 75% ethanol was used as blank control and 500 μmol/L 90% ascorbic acid (ASA) methanol solution was used as the positive control. All experiments were performed in triplicate.

### 2.7. Microstructure

The method of Huang et al. [22] was slightly changed, and the samples were observed by scanning electron microscope (S-4800N, Hitachi Corporation, Japan). Firstly, materials were cut into small slices of 5 mm × 5 mm and quickly immersed in 2.5% glutaraldehyde solution for 12 h. Then, rinsed 3 times with 0.2 mol/L pH 7.4 phosphate buffer, 15 min each time. Gradient elution with 50%, 70%, 80%, 90%, and 100% ethanol for 15 min each time. Finally, the infiltrated samples were transferred to tert-butanol for preservation, and the microstructure of the samples were observed after gold spray treatment.

### 2.8. Principal Components Analysis (PCA)

PCA is a statistical method that is applied for the analysis and dimension reduction of complex data, more specifically, to identify the redundant variables that do not add new information but complicate the data. It allows the conversion of relevant variables into a set of ordered, uncorrelated variables, known as principal components [40]. Areas of physicochemical qualities were subjected to PCA and Pearson analysis. Scores obtained from each PCA were analyzed to a one-way analysis of variance (ANOVA) to test for significant differences between samples [41].

### 2.9. Entropy Weight-Coefficient of Variation Method Coupling Weight

The optimum drying conditions of *Angelica sinensis* were determined by entropy weight-coefficient of variation comprehensive scoring method. The calculation of index weight and comprehensive score were calculated as in Equations (7) and (8):(5)$$A_{\omega 1}=\frac{\alpha }{\sum_{i=1}^{m}\alpha }$$
(6)$$A_{\omega 2}=\frac{\beta_{i}}{\sum_{i=1}^{m}\beta_{i}}$$
(7)$$A_{\omega }=\frac{\sqrt{A_{\omega 1\cdot A_{\omega 2}}}}{\sum_{i=1}^{m}\sqrt{A_{\omega 1}\cdot A_{\omega 2}}}$$
(8)$$Y=\sum_{i=1}^{m}\frac{100\times A_{\omega }\times X_{j}}{X_{imax}}$$
where *A_ω_*_1_ is the weight value of entropy weight method; *α* and *β* are the coefficient of variation; *A_ω_*_2_ is the weight value of variation coefficient method; *A_ω_* is the optimized coupling weights by Lagrange multiplier method; *Y* is the comprehensive scoring.

### 2.10. Statistical Analysis

The data are presented as the mean of three determinations ± standard deviation. The data were analyzed by ANOVA and Duncan’s multiple-range test using SPSS statistics software (Version 22.0, IBM SPSS Digital Analytics, Co., Ltd, New York, NY, USA). Statistical significance for differences was tested at 5% probability level (*p* < 0.05). To ensure the reliability of the test, all tests were repeated three times and the average value was taken as the test value.

## 3. Results and Discussion

### 3.1. Drying Characteristics

#### 3.1.1. Effect of Radiation Temperatures on the Drying Characteristics

When the slice thickness was 4 mm and the vacuum degree was −70 kPa, the drying characteristic curve of *Angelica sinensis* at different radiation temperatures were shown in Figure 2. With increasing radiation temperature, the time to reach safe moisture content in the *Angelica sinensis* gradually decreased. At the stage of high moisture content, most of the water inside belongs to free water and unbound water, which is conducive to the propagation and penetration of microwave inside the material. As drying proceeds, the trend of decreasing moisture content gradually slows down and the drying rate decreases. This may be due to the attenuation coefficient of microwave propagating in liquid is much smaller than that in solid [32]. Compared with the samples control, the drying time was reduced by 40.0%, 53.3%, 60.0%, 60.01%, and 60.8% after the microwave vacuum drying was applied for 35 °C, 40 °C, 45 °C, 50 °C min, and 55 °C, respectively. This is because the vacuum drying allows water to vaporize at lower temperatures than at atmospheric pressure, and the increase in temperature enhances the vapor pressure difference between the material and the drying medium, improves the internal energy and activity of water molecules, promotes the evaporation and diffusion of water, accelerates the heat transfer rate, and shortens the drying time. Similar results were also found in ginger [42] and Stevia [22] after microwave vacuum treatment. Secondly, under the high microwave output power heating, the material absorbs microwave energy, making the intermolecular forces of bound water weaken, destroying the internal chemical bonds, and transforming part of the bound water in the cell into free water with better mobility, which reduced the internal diffusion boundary and enhanced the heat and mass transfer efficiency. In addition, when the drying temperature was 45~55 °C, the total drying time was not much different, which may be due to the high temperature creating sample shrinkage and a surface hardening phenomenon, increasing the resistance of internal moisture transfer and reducing the evaporation of water. At the same time, the drying rate showed a trend of first increasing and then decreasing. This is because the pressure difference between inside and outside the material increases under the action of microwave electric field, which improves the drying rate. In the later stage of drying, the material shrinkage was more serious, the internal permeability becomes lower, and the drying rate decreases.

#### 3.1.2. Effect of Slice Thickness on Drying Characteristics

When the drying temperature was 45 °C and the vacuum degree was −70 kPa, the drying characteristic curves of *Angelica sinensis* at different slice thickness were shown in Figure 3. The curves showing the moisture ratio change in samples almost coincide. This shows that the slice thickness has little effect on the process of drying *Angelica sinensis* with microwave vacuum drying, because the microwave penetration depth may be greater than 6 mm, and the microwave energy was absorbed by the material almost the same, so the drying characteristic curve of the material changes less. In addition, the drying rate gradually decreased and constant-rate stage could not be observed and the microwave vacuum drying of *Angelica sinensis* was a falling-rate drying period, indicating that the drying process was mainly controlled by internal diffusion. At this time, the internal water diffusion rate is lower than the surface evaporation rate. At the early stage of drying, the free water in the material was high and easy to remove, so the drying rate was faster. With the evaporation of free water, the water in the material was mainly bound water, the water migration rate slows down, resulting in the absorption of microwave energy was reduced, the internal and external vapor pressure difference was degraded accordingly, and the drying rate was decreased.

#### 3.1.3. Effect of Vacuum Degree on the Drying Characteristics

When the drying temperature 45 °C and slice thickness was 4 mm, the drying characteristic curve of *Angelica sinensis* at different vacuum degrees were shown in Figure 4. The drying time tends to decrease first and then increase as the vacuum level decreases. This may be because the decrease of vacuum degree leads to the increase of total pressure difference inside and outside the sample, and the interaction of bipolar particles inside the material under the action of electromagnetic field makes the molecules moving, friction and generation heat, and this process rapidly accumulates energy and internal pressure, therefore, the rate of moisture migration to the material surface is accelerated [43,44]. Normally, as the vacuum decreases, the air pressure inside the sealed chamber decreases, the saturation vapor pressure of water also reduces, and the water was more easily removed from the material under the same heating conditions, but when the vacuum was −80 kPa, the drying rate decreases, and the drying time increases [45].

### 3.2. Drying Kinetics

To study the material drying kinetics, the experimental moisture ratio and the predicted moisture ratio using regression analysis are shown in Figure 5. The slope and intercept of the regression curve closer to 1 and 0 represent better prediction accuracy, respectively. As shown in Figure 5, the R^2^ of the Newton (Figure 5A), Handerson (Figure 5D) and Two-term exponential (Figure 5F) models ranged from 0.99388 to 0.99608, the coefficient of determination of Pearson’s ranged from 0.99795 to 0.99841. In comparison, the R^2^ of Midilli (Figure 5B), Weibull (Figure 5C), and Logarithmic (Figure 5E) were 0.99903, 0.99768, and 0.99738, respectively, and the slopes were 0.99932, 1.00538, and 0.99741, respectively, so these three models were more accurate in predicting the experimental moisture ratios, with Milldi model was the best. This further provided a theoretical basis for describing the drying characteristics of *Angelica sinensis*.

### 3.3. Color

Color is an external indicator to measure the retention degree of nutritional components in food [36]. The color difference (ΔE) was usually used as one of the comprehensive indexes to evaluate the color change of materials. The smaller ΔE means the lower browning and the better the quality of the food. The pattern of changes in color parameters of *Angelica sinensis* under different drying conditions can be seen in Table 2. The *L*_0_, *a*_0_, and *b*_0_ values of fresh samples were 80.98 ± 4.32, 0.40 ± 0.23, and 14.51 ± 1.89, respectively. Compared with the control samples, the *L** value increased, the *a** and *b** values reduced, and the ∆E value decreased after microwave vacuum drying. The corresponding *L** value at 35 °C is 11.88% lower than that at 45 °C. This may be due to the lower drying temperature and the prolonged contact of the material with wet air, which increases the probability of oxidation reaction, resulting in a significant decrease in brightness value [46]. When the temperature rises to 55 °C, the highest ΔE (18.90 ± 2.43) and the lowest *L** (63.58 ± 1.32) were obtained. There was a significant negative relationship between ΔE and *L** (r = 0.93, *p* < 0.01). This is because high temperatures lead to phenols, glycosides and other substances degradation, (e.g., hydroxyl aldehyde condensation and polymerization), exacerbated by the Maillard reaction and caramelization reaction, making the *L** value was reduced [47]. At 45 °C, the ΔE (6.77 ± 2.01) of the dried product was lower, indicating that the color of the sample was closer to the fresh sample under this condition. The lower color change could be related to the improvement of heating uniformity from tray rotation [48]. Color difference and drying characteristics showed a similar trend, probably because the color change and drying time is related. Combining the effects of different drying conditions on the color of *Angelica sinensis*, the lowest ΔE and the closest color to the fresh sample were obtained when drying temperature was 45 °C, slice thickness was 3 mm, and vacuum degree was −65 kPa.

### 3.4. Effect on Phytochemical Quality of Angelica Sinensis

#### 3.4.1. Natural Active Ingredient Content

Organic acids and phthalides are the main natural active ingredient of *Angelica sinensis*, which are the key indexes affecting the sensory properties and nutrition value of the product, including chlorogenic acid, ferulic acid, ligustilide, senkyunolide H, senkyunolide I, and 3-butenylphthalide, etc. It has antioxidant, hypolipidemic, anti-atherosclerosis, antibacterial, anti-inflammatory, and enhanced immunomodulatory effects [2,3]. The contents of natural active ingredients of the samples under different drying condition is shown in Table 3 and Figure 6. The highest contents of ferulic acid (1676.22 ± 9.98 μg/g), senkyunolide I (168.73 ± 1.29 μg/g), senkyunolide H (278.92 ± 3.59 μg/g), and ligustilide (2520.45 ± 10.22 μg/g) were obtained at a drying temperature of 45 °C, slice thickness of 4 mm and vacuum of −70 kPa. The content of chlorogenic acid (860.10 ± 9.24 μg/g) was the highest when the drying temperature was 50 °C. Additionally, different drying conditions also had a great influence on the content of each active ingredient. For example, when the radiation temperature increased from 35 °C to 50 °C, the content of ferulic acid increased by 90.64%; when the vacuum degree increased from −60 kPa to −75 kPa, the contents of chlorogenic acid, senkyunolide I, and ligustilide increased first and then decreased. This may be due to the oxidative damage of the cell wall caused by higher vacuum degree, which resulted in some peroxidases leaking out from the cytoplasmic matrix, thus leading to the decrease in content [32].

#### 3.4.2. Total Phenol and Total Flavonoid Content

Phenols and flavonoids are heat-sensitive components with high activity and antioxidant activity [37,38]. The contents of total phenols and total flavonoids in samples under different drying conditions ranged from 145.70 to 190.82 mg/100 g and 174.84 to 274.39 mg/100 g. The total phenol content increased by 18.86% to 55.82% after microwave vacuum drying compared to the sample control. As shown in Figure 7, drying temperature, slice thickness, and vacuum degree had significant effects on the contents of phenols and flavonoids (*p* < 0.05). When the drying temperature was 50 °C, the total phenolic content of dried products (190.82 mg/100 g) was the highest. The increase in radiation temperature also caused an increase in TFC was found in Parveez et al. [24]. At 35 °C, the total phenolic content of dried products (145.70 mg/100 g) was the lowest. This may be due to the lower temperature and longer drying time, and phenolic compounds were prone to oxidative degradation under the catalysis of enzymes, which reduces the content of total phenolic compounds [16,49]. At the same time, it can be seen that the total phenol and total flavonoid contents of the samples showed a trend of increasing and then decreasing with the increase of section thickness, and the highest total phenol content (190.82 mg/100 g) and total flavonoid content (253.12 mg/100 g) were found at 4 mm, which indicates that the microwave vacuum drying under this condition has less damage to the cellular structure of the material, which makes the internal polar molecules increase and retains its medicinal value better. Furthermore, the vacuum degree also showed the same trend as the slice thickness, and the maximum content of total phenols and total flavonoids was reached at the vacuum degree of −70 kPa. Collectively, the highest retention of total phenols and total flavonoids in the samples was achieved when the drying temperature was 50 °C, the slice thickness was 4 mm, and the vacuum degree was −70 kPa.

#### 3.4.3. Antioxidant Capacity

Antioxidant capacity refers to the inhibition of oxidative chain reactions by providing hydrogen donors or free radical acceptors through antioxidants such as phenolic compounds. The multi-antioxidant properties of samples under different drying conditions are shown in Figure 7. The effects of conditions factors on the antioxidant capacity were statistically significant (*p* < 0.05). The antioxidant capacity in the materials after microwave vacuum drying was higher than the 15.07% observed in the control. The free radical scavenging rates at 40 °C, 45 °C, and 50 °C were 55.27%, 60.29%, and 58.79%, respectively, indicating that increasing the drying temperature appropriately was beneficial to improve the antioxidant capacity. Similar results were achieved by Zannou et al. [50], Nguyen et al. [43], and Zhang et al. [51]. This was because microwave vacuum drying could highly maintain the biological activities of phenols and flavonoids, Ozan et al. [9] also reached a similar conclusion. The Pearson’s correlation analysis results revealed that the antioxidant capacity exhibited significant positive correlations with ligustilide (*p* < 0.05). Similarly, the antioxidant capacity also displayed strong positive correlations with total phenolics (*p* < 0.01) and total flavonoids (*p* < 0.01). The ligustilide, flavonoids, and phthalide in the volatile oil of *Angelica sinensis* also have certain antioxidant properties, which enhance its antioxidant capacity to some extent. As shown in Figure 7, the antioxidant capacity of the material was the strongest (60.29%) when the drying temperature was 45 °C, the slice thickness was 4 mm, and the vacuum degree was −70 kPa. Compared with other drying conditions, the above conditions promoted the inactivation of oxidase and peroxidase in the samples to some extent, thereby reducing the degradation of phenolic and ketone substances.

#### 3.4.4. Polysaccharide Content

The polysaccharide content of samples under different drying conditions is shown in Figure 8. With the increase in drying temperature, the polysaccharide content increased first and then decreased. The polysaccharide content was 179.92 mg/g at the drying temperature of 50 °C, which increased by 42.84% compared with that at 35 °C (125.96 mg/g). This may be due to the low drying temperature, which increased the time for the Maillard reaction and caramelization reaction between amino acids and polysaccharides, and therefore the polysaccharide content decreased [37,39]. It is also known that the polysaccharide contents of the samples were 140.96 mg/g and 141.80 mg/g at section thicknesses of 3 mm and 4 mm, which were higher than the polysaccharide contents at other thicknesses, indicating that the samples could retain the bioactive substances better under this condition. Different vacuum degrees also had significant effects on polysaccharide content (*p* < 0.05), and the highest polysaccharide content was found when the vacuum degree was −75 kPa. This is because *Angelica sinensis* contains a large amount of starch, and its amylopectin is prone to de-chaining under the action of electromagnetic field, forming amylose that is easily soluble in water. The lower the vacuum degree, the greater the internal pressure, and the weaker the decomposition of starch, thus the loss of polysaccharides was less.

### 3.5. Microstructure

The microstructure of the material determines the macroscopic properties, and its internal moisture diffusion and heat mass transfer rate are closely related to the change of microstructure. The macroscopic and microscopic structures of the samples under different drying conditions are shown in Figure 9. As shown in Figure 9B, the internal cells of fresh *Angelica sinensis* were closely arranged, while the microstructure of cell wall and cell membrane in the dried samples changed significantly. After microwave drying (Figure 9D), the surface organization of the material was dense, with fewer microporous channels, and the appearance of obvious shrinkage and deformation, which is not conducive to the diffusion and migration of water [52]. Compared with the control group, the internal pores of the sample increased significantly after microwave vacuum drying (Figure 9H). This may be because the sample absorbed a large amount of microwave energy at this time, increasing in the internal and external vapor pressure difference, improving the fluidity of water molecules, and causing the tissue cells to expand rapidly [53]. The increase of internal pores leads to the formation of a regular honeycomb pore structure. He et al. [54] and others also reached similar conclusions. As shown in Figure 9F, the higher drying temperature causes severe shrinkage deformation and fracture collapse of internal tissue cells, which may be due to the rapid loss of moisture and rapid decrease of expansion pressure inside the sample caused by high temperature, surface hardening phenomenon, strong local stress was generated, and the organizational structure was destroyed, which is consistent with the analysis results of drying characteristics. The effects of vacuum degree and slice thickness on the microstructure also showed similar results, thus the appropriate vacuum degree and slice thickness could inhibit the contraction of tissue cells and the collapse of the structure, to speed up the internal water migration efficiency and reduce the heat and mass transfer resistance has a positive effect.

### 3.6. PCA Analysis and Pearson’s Correlation Analysis

To investigate the overall effect of microwave vacuum drying process on drying characteristics and physicochemical quality, principal component analysis (PCA) was applied to the whole data, and Figure 10A. shows the PCA of physicochemical quality. Among them, PC1 and PC2 were 52.1% and 20.5%, respectively, which were sufficient to explain the total variance in the dataset. In the PC1, total phenols, ferulic acid, chlorogenic acid, total flavonoids, senkyunolide I, antioxidant activity were strongly positively correlated, consistent with the Pearson correlation. On the other hand, ΔE and b* were negatively correlated. The PCA of different drying processes as shown in Figure 10A, 3 and Fresh *Angelica sinensis* were found on the positive side of PC1 and close to each other, indicating that there may be similarities between the three and the fresh samples. This confirms that 3 (45 °C, 4 mm, −70 kPa) can better retain the nutrients in the sample, and is consistent with the entropy-coefficient of variation comprehensive score results (Figure 11). Figure 10B shows the Pearson’s correlation analysis between different physicochemical qualities, which allows a better understanding and analysis of the relationship between the variables. There was a highly significant positive correlation between total flavonoids and total phenols, probably because total phenols and total flavonoids are heat-sensitive components with synergistic effects [55]. The polysaccharide exhibited antagonistic effects with ligustilide and senkyunolide H. Studies have shown that TPC associates with antioxidant activity, Similar results have been found by many researchers, namely, Lim [56] and Khokha [57].

### 3.7. Entropy Weight-Coefficient of Variance Composite Score

The entropy weight-variation coefficient method evaluates the influence degree of each index on the whole by calculating the coupling weight coefficient, which effectively avoids the shortcomings of unreasonable distribution of single objective weight, and reflects the actual weight of each index more objectively than the previous subjective assignment method, making the evaluation results more coordinated, compatible, and scientific. Drying time, color difference, antioxidant activity, polysaccharides, total phenols, total flavonoids, chlorogenic acid, ferulic acid, senkyunolide H, senkyunolide I, butenylphthalide, and ligustilide were selected as evaluation indexes, and ligustrolactone were selected as the evaluation indexes, and the weight coefficients Aω of the comprehensive evaluation indexes were obtained according to the entropy weight-variance coefficient method. The results showed that the drying quality of *Angelica sinensis* was the best (comprehensive score was 84.66) when the drying temperature was 45 °C, the slice thickness was 4 mm, and the vacuum degree was −70 kPa (Figure 11).

## 4. Conclusions

In this research, drying characteristics, physicochemical qualities, and microstructures of *Angelica sinensis* slices dried with microwave vacuum drying were examined. The results showed that the higher drying temperature and vacuum degree, smaller slice thickness, the shorter the drying time and the faster the drying rate, which indicated that microwave vacuum drying could effectively enhance the heat and mass transfer efficiency. Comparing the physicochemical quality of *Angelica sinensis* slices under various factors, it was found that drying temperature, slice thickness, and vacuum degree had significant effects on the color, natural active ingredients and essential nutrients of the material. When the drying temperature was 45 °C, the slice thickness was 4 mm, and the vacuum degree was −70 kPa, the physicochemical qualities were the highest in the dried products, and entropy weight-coefficient of variation method highest comprehensive score (84.66). The Midilli model best fitted the experimental results for drying with the coefficient compared with the other models. Therefore, Midilli model could describe the *Angelica sinensis* internal water mass transfer characteristics by predicting the moisture ratio under different drying conditions. Comparing the macro and micro structures of *Angelica sinensis* after different drying methods, it was found that the uniform and regular honeycomb-like pore structure appeared inside the material after microwave vacuum drying, which effectively reduced the structural damage and had the best color (ΔE = 6.77 ± 2.01). This study provides a useful exploration and practice for the formation of drying processing technology of rhizome medicinal materials, and has the potential to be applied on a commercial scale.

## Figures and Tables

**Figure 1 foods-12-01202-f001:**
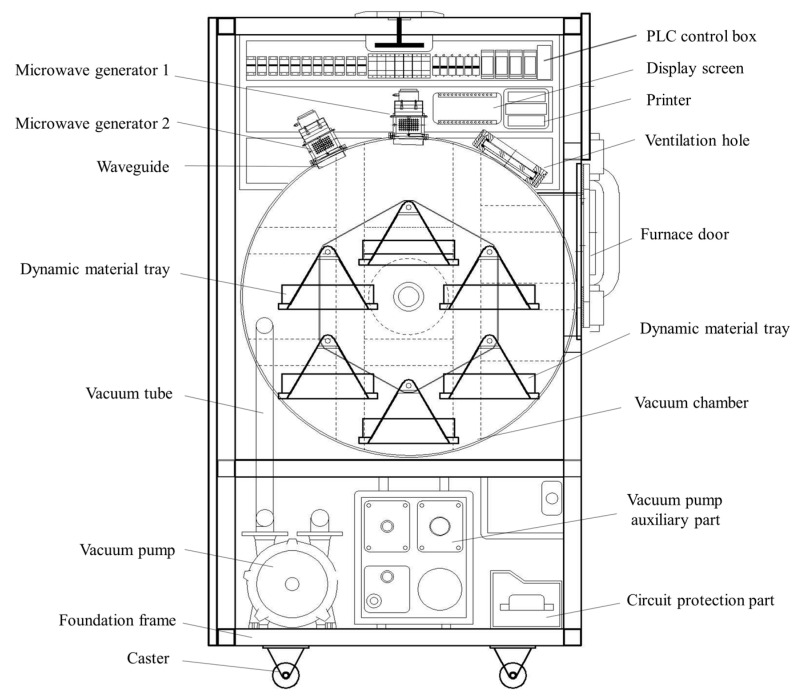
Schematic diagram of the tray rotation microwave vacuum dryer.

**Figure 2 foods-12-01202-f002:**
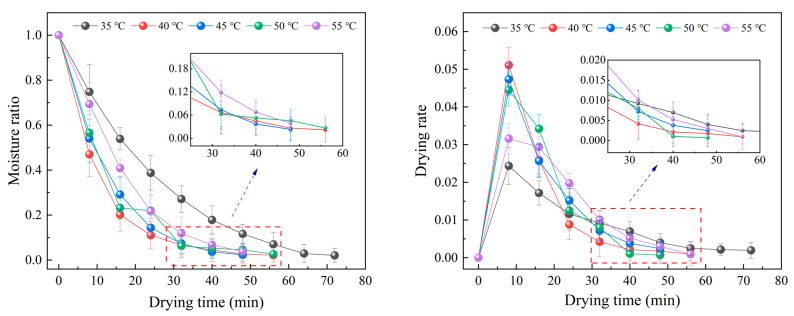
Effects of radiation temperature on drying curve and drying rate curve.

**Figure 3 foods-12-01202-f003:**
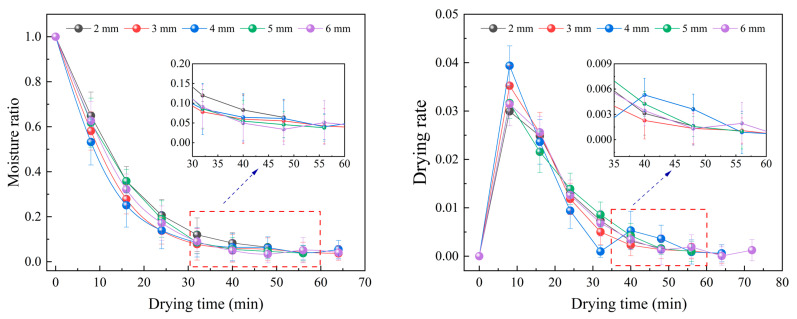
Effects of slice thickness on drying curve and drying rate curve.

**Figure 4 foods-12-01202-f004:**
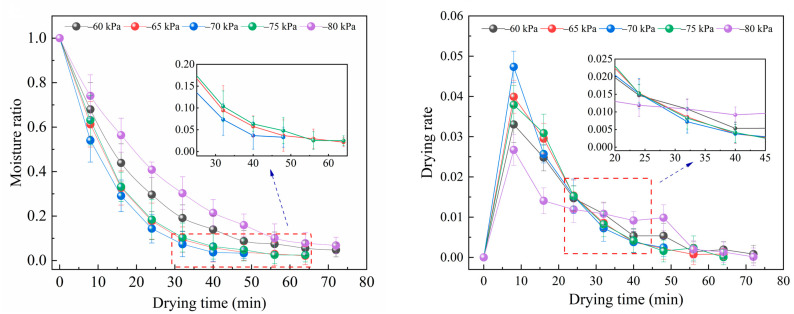
Effects of vacuum degree on drying curve and drying rate curve.

**Figure 5 foods-12-01202-f005:**
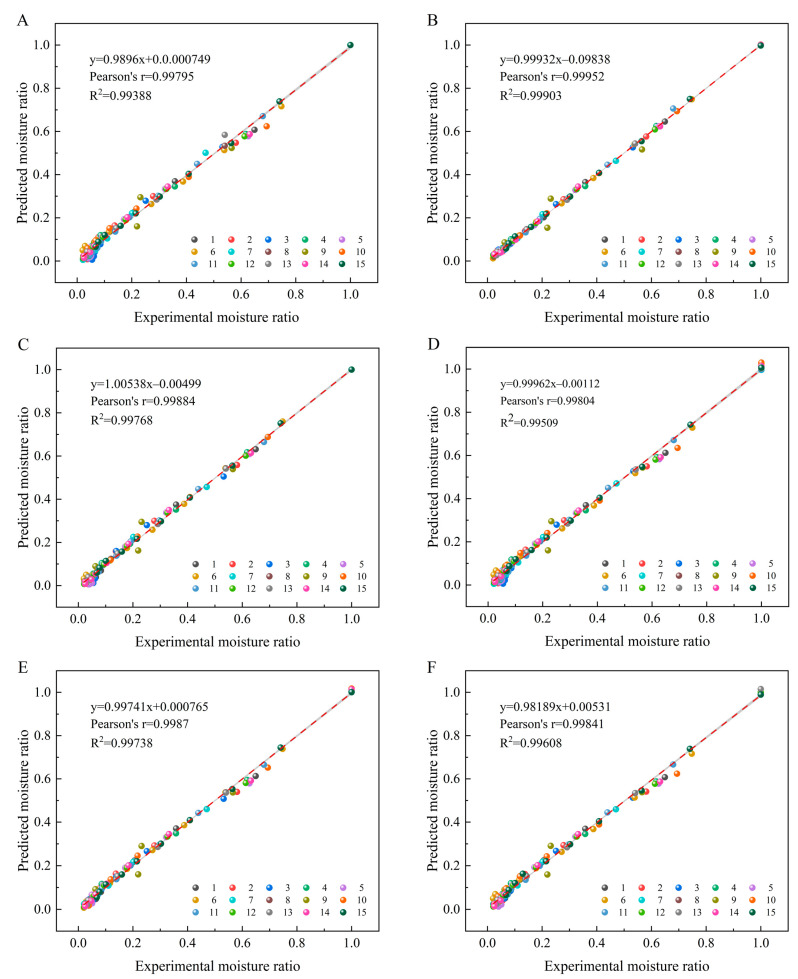
Experimental determined and predicted moisture ratio by Newton (**A**), Midilli (**B**), Weibull (**C**), Handerson and Pabis (**D**), Logarithmic (**E**) and Two-term exponential (**F**) model with different drying conditions.

**Figure 6 foods-12-01202-f006:**
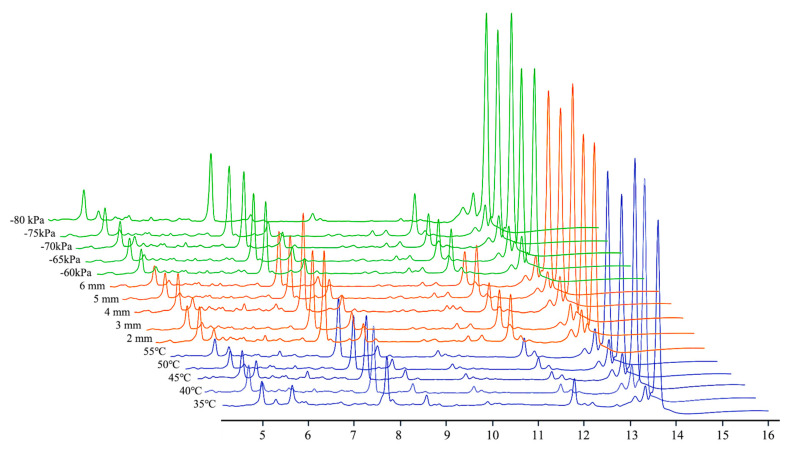
HPLC chromatograms of the natural active ingredient contents under different drying conditions.

**Figure 7 foods-12-01202-f007:**
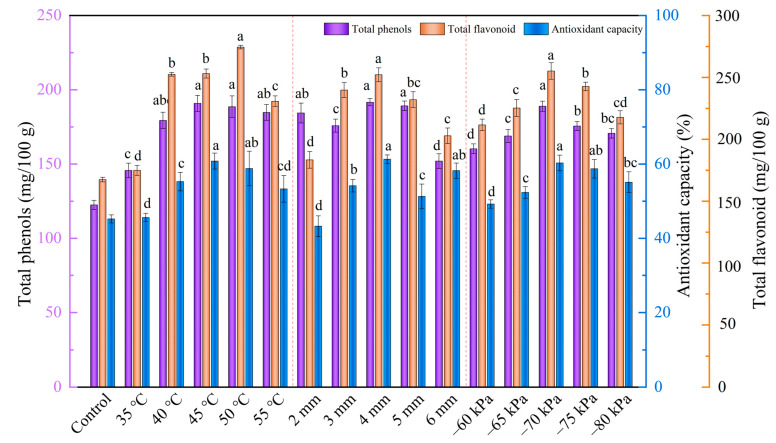
Total phenol, total flavonoid, and antioxidant capacity of *Angelica sinensis* dried before and after microwave vacuum drying.

**Figure 8 foods-12-01202-f008:**
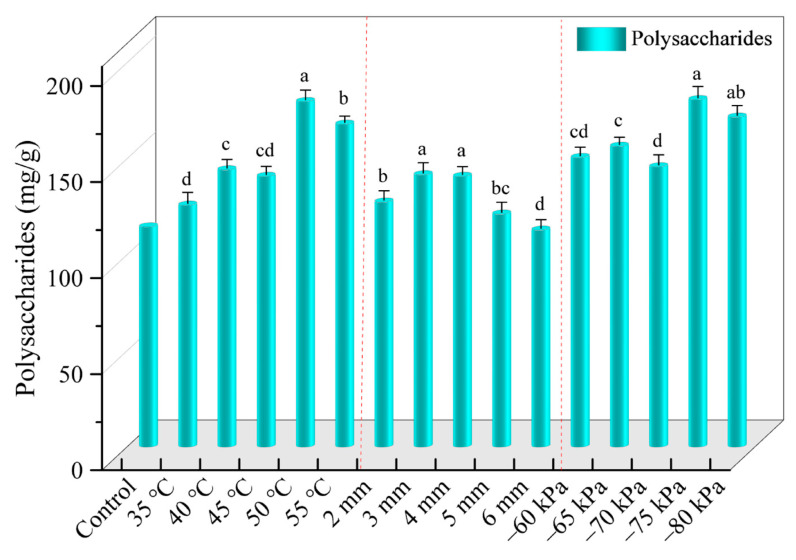
Polysaccharides of *Angelica sinensis* dried before and after microwave vacuum drying.

**Figure 9 foods-12-01202-f009:**
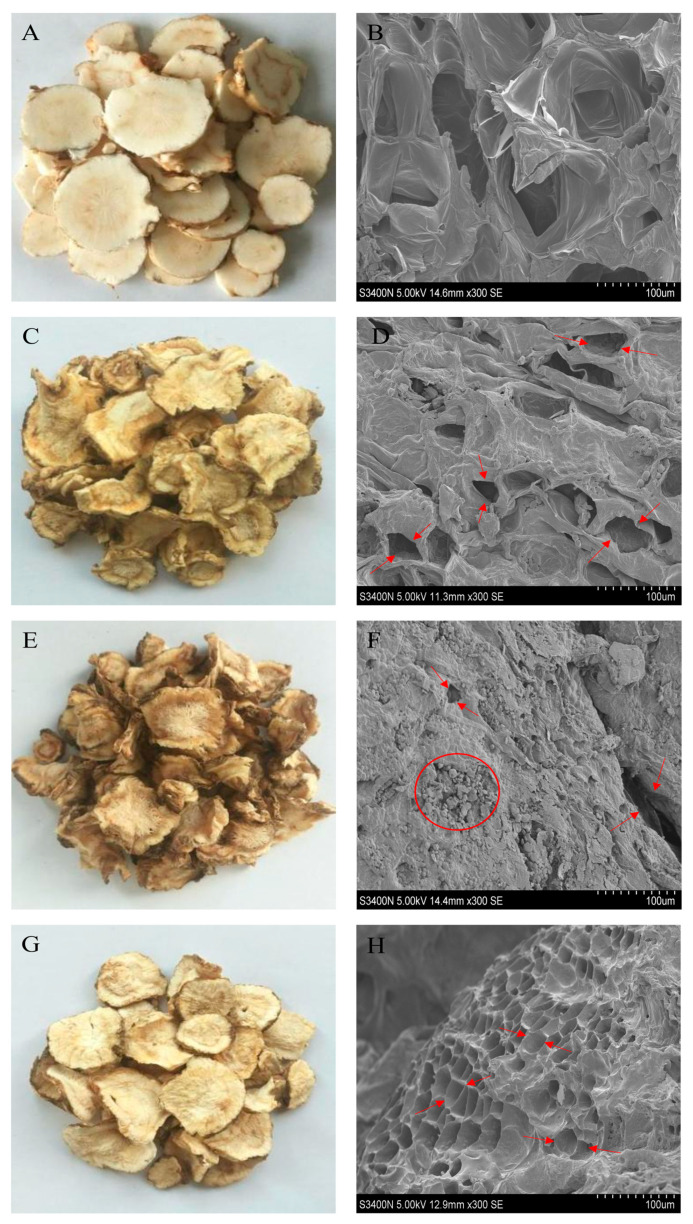
Macroscopic and microscopic of *Angelica sinensis* under different drying conditions. (**A**,**B**) Fresh samples (**C**,**D**) Microwave drying (**E**,**F**) 55 °C/4 mm/−70 kPa (**G**,**H**) 45 °C/4 mm/−70 kPa.

**Figure 10 foods-12-01202-f010:**
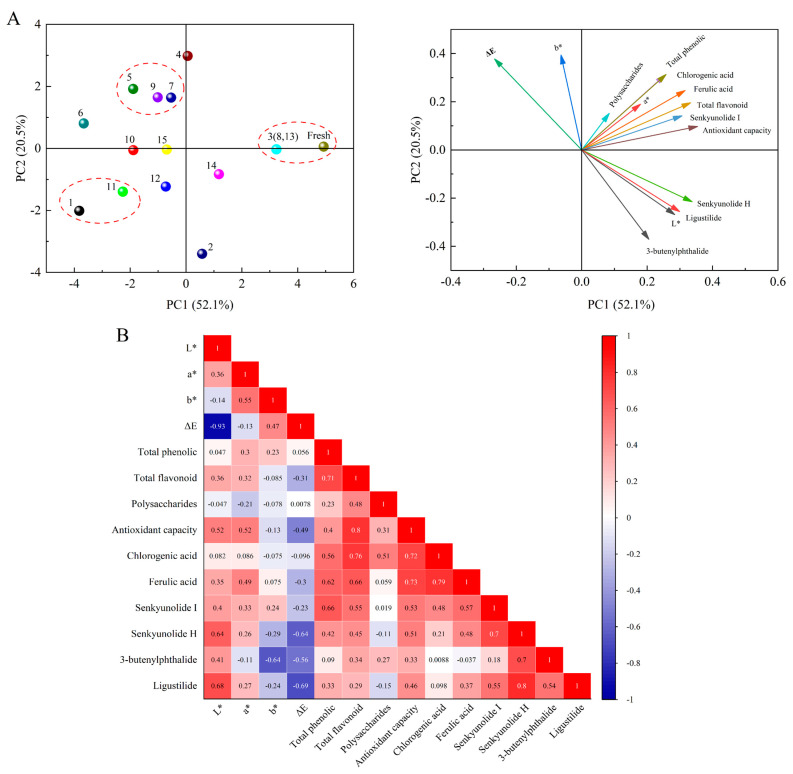
Principal component analysis (**A**) and Pearson correlation analysis (**B**) plot of physicochemical quality obtained from different drying treatments.

**Figure 11 foods-12-01202-f011:**
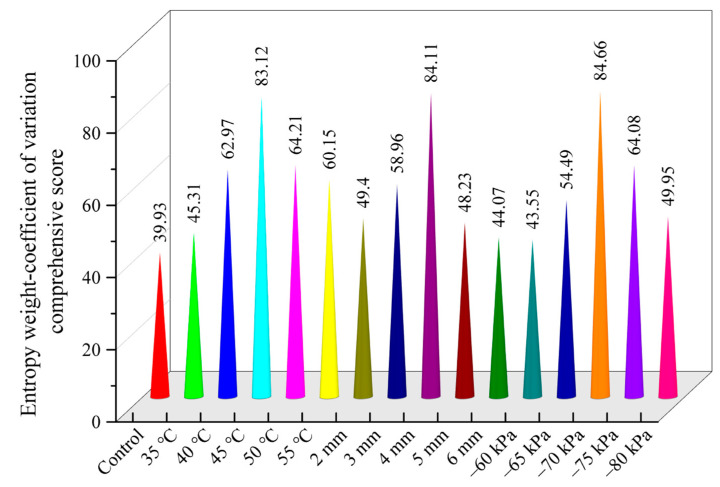
The results of entropy weight-coefficient of variation comprehensive score under different drying conditions.

**Table 1 foods-12-01202-t001:** Drying models used for drying kinetics of *Angelica sinensis*.

Model Name	Mathematical Equation
Newton	MR = exp(−kt)
Midilli	MR = aexp(−kt^n^) + b
Weibull	MR = exp[−(k/α)^β^]
Handerson and Pabis	MR = aexp(−kt)
Logarithmic	MR = aexp(−kt) + b
Two-term exponential	MR = aexp(−kt) + (1 − a)exp(−kat)

**Table 2 foods-12-01202-t002:** Color of *Angelica sinensis* dried before and after microwave vacuum drying.

Experiments Number	Drying Conditions	Color			
		*L**	*a**	*b**	ΔE
1	35 °C/4 mm/−70 kPa	69.69 ± 2.59 ^bcd^	1.77 ± 0.89 ^e^	19.01 ± 2.12 ^cd^	12.38 ± 3.46 ^bc^
2	40 °C/4 mm/−70 kPa	74.21 ± 1.08 ^b^	1.91 ± 0.32 ^de^	15.38 ± 1.06 ^e^	6.99 ± 1.64 ^de^
3	45 °C/4 mm/−70 kPa	77.67 ± 0.89 ^a^	2.84 ± 1.01 ^bc^	20.10 ± 2.01 ^c^	6.70 ± 1.98 ^de^
4	50 °C/4 mm/−70 kPa	66.54 ± 2.01 ^d^	1.97 ± 1.12 ^de^	19.82 ± 3.12 ^cd^	15.54 ± 3.41 ^b^
5	55 °C/4 mm/−70 kPa	63.58 ± 1.32 ^e^	2.28 ± 0.67 ^d^	21.63 ± 2.32 ^bc^	18.90 ± 2.43 ^a^
6	45 °C/2 mm/−70 kPa	64.92 ± 2.61 ^de^	1.81 ± 0.61 ^e^	22.01 ± 1.11 ^b^	17.85 ± 1.98 ^ab^
7	45 °C/3 mm/−70 kPa	76.96 ± 1.91 ^ab^	3.59 ± 1.21 ^a^	25.21 ± 0.89 ^a^	11.87 ± 1.79 ^bc^
8	45 °C/4 mm/−70 kPa	77.77 ± 1.77 ^a^	2.81 ± 1.23 ^bc^	20.22 ± 1.71 ^c^	6.79 ± 2.58 ^de^
9	45 °C/5 mm/−70 kPa	71.48 ± 2.09 ^c^	3.42 ± 0.29 ^ab^	21.76 ± 0.98 ^bc^	12.36 ± 1.95 ^bc^
10	45 °C/6 mm/−70 kPa	71.93 ± 2.22 ^c^	3.10 ± 2.01 ^b^	19.61 ± 1.21 ^cd^	10.75 ± 3.01 ^c^
11	45 °C/4 mm/−60 kPa	73.81 ± 1.77 ^bc^	1.36 ± 0.99 ^f^	18.96 ± 0.66 ^d^	8.98 ± 1.44 ^d^
12	45 °C/4 mm/−65 kPa	77.18 ± 3.01 ^a^	1.65 ± 0.37 ^ef^	18.62 ± 1.19 ^d^	6.07 ± 2.05 ^e^
13	45 °C/4 mm/−70 kPa	77.97 ± 1.89 ^a^	2.86 ± 1.01 ^bc^	20.04 ± 2.59 ^c^	6.77 ± 2.01 ^de^
14	45 °C/4 mm/−75 kPa	74.93 ± 2.31 ^b^	2.96 ± 0.79 ^bc^	19.62 ± 2.36 ^cd^	8.50 ± 2.65 ^d^
15	45 °C/4 mm/−80 kPa	74.93 ± 1.95 ^b^	2.62 ± 0.22 ^c^	20.75 ± 3.45 ^c^	8.53 ± 3.21 ^d^

Note: Data are expressed as means ± standard deviation of triplicate samples. The letters reveal significant differences (*p* < 0.05) according to the Duncan test.

**Table 3 foods-12-01202-t003:** The content of natural active ingredients in *Angelica sinensis* dried before and after microwave vacuum drying.

Drying Conditions	Organic Acids/(μg/g)		Phthalides/(μg/g)			
	Chlorogenic Acid	Ferulic Acid	Senkyunolide I	Senkyunolide H	3-Butenylphthalide	Ligustilide
Control	216.48 ± 6.77 ^cde^	849.62 ± 4.47 ^f^	24.46 ± 4.91 ^def^	189 ± 3.35 ^bc^	211.67 ± 4.72 ^g^	1620.52 ± 8.49 ^e^
35 °C/4 mm/−70 kPa	132.60 ± 7.29 ^e^	857.08 ± 5.74 ^f^	19.41 ± 0.37 ^f^	153.25 ± 6.49 ^c^	403.29 ± 4.27 ^cd^	1847.70 ± 12.01 ^d^
40 °C/4 mm/−70 kPa	246.84 ± 4.46 ^d^	888.22 ± 7.47 ^ef^	49.46 ± 4.57 ^de^	235.68 ± 2.17 ^b^	712.55 ± 7.32 ^a^	2341.44 ± 9.01 ^b^
45 °C/4 mm/−70 kPa	579.10 ± 6.29 ^b^	1681.90 ± 5.71 ^a^	167.07 ± 4.24 ^a^	281.92 ± 3.59 ^a^	450.81 ± 7.62 ^c^	2528.22 ± 8.09 ^a^
50 °C/4 mm/−70 kPa	860.10 ± 9.24 ^a^	1633.94 ± 4.38 ^ab^	69.04 ± 3.11 ^d^	74.95 ± 2.01 ^e^	242.39 ± 1.29 ^ef^	1611.70 ± 13.21 ^e^
55 °C/4 mm/−70 kPa	457.92 ± 3.99 ^c^	986.25 ± 5.49 ^cde^	89.64 ± 2.08 ^c^	88.44 ± 0.37 ^de^	346.24 ± 6.21 ^d^	1806.88 ± 12.01 ^de^
45 °C/2 mm/−70 kPa	199.49 ± 5.08 ^de^	949.52 ± 6.99 ^e^	62.76 ± 1.49 ^d^	108.25 ± 0.99 ^d^	228.62 ± 2.04 ^f^	2016.40 ± 14.22 ^c^
45 °C/3 mm/−70 kPa	230.79 ± 6.55 ^d^	1022.86 ± 9.23 ^de^	118.55 ± 4.66 ^b^	132.21 ± 4.22 ^cd^	271.35 ± 7.44 ^e^	1954.44 ± 6.28 ^bcd^
45 °C/4 mm/−70 kPa	582.37 ± 8.29 ^b^	1676.22 ± 9.98 ^a^	168.73 ± 1.29 ^a^	278.92 ± 3.59 ^a^	446.88 ± 5.62 ^c^	2520.45 ± 10.22 ^a^
45 °C/5 mm/−70 kPa	330.26 ± 5.11 ^cd^	1520.40 ± 8.97 ^b^	16.13 ± 1.29 ^f^	116.49 ± 2.01 ^d^	211.95 ± 2.49 ^g^	1983.60 ± 13.22 ^cd^
45 °C/6 mm/−70 kPa	347.09 ± 7.63 ^c^	1255.23 ± 4.59 ^cd^	14.81 ± 0.42 ^f^	86.43 ± 3.66 ^de^	229.56 ± 7.42 ^f^	1997.91 ± 11.88 ^cd^
45 °C/4 mm/−60 kPa	359.73 ± 3.66 ^c^	896.28 ± 3.11 ^ef^	14.69 ± 3.12 ^f^	66.38 ± 2.33 ^e^	324.78 ± 6.72 ^de^	2166.12 ± 7.01 ^bc^
45 °C/4 mm/−65 kPa	401.22 ± 4.37 ^bc^	1158.18 ± 3.99 ^d^	51.64 ± 0.91 ^de^	146.28 ± 3.01 ^cd^	384.86 ± 2.97 ^cd^	1845.26 ± 15.01 ^d^
45 °C/4 mm/−70 kPa	588.37 ± 10.01 ^b^	1671.31 ± 7.68 ^a^	166.13 ± 1.22 ^a^	284.32 ± 5.79 ^a^	449.78 ± 3.42 ^c^	2516.30 ± 11.47 ^a^
45 °C/4 mm/−75 kPa	461.13 ± 5.16 ^c^	1306.10 ± 7.71 ^c^	32.61 ± 3.10 ^e^	199.24 ± 6.48 ^bc^	624.19 ± 4.99 ^b^	2200.65 ± 13.22 ^bc^
45 °C/4 mm/−80 kPa	359.35 ± 6.57 ^c^	1142.94 ± 10.21^c^	34.56 ± 1.49 ^e^	96.93 ± 3.66 ^cde^	326.28 ± 3.84 ^de^	2197.82 ± 8.10 ^bc^

Note: Data are expressed as means ± standard deviation of triplicate samples. The letters reveal significant differences (*p* < 0.05) according to the Duncan test.

## Data Availability

Data is contained within the article.

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
