# Peer review of "Improving Drying Characteristics and Physicochemical Quality of Angelica sinensis by Novel Tray Rotation Microwave Vacuum Drying"

_foods, 2023, doi:10.3390/foods12061202_

Round 1
Reviewer 1 Report
The manuscript received for review presents the research aiming to evaluate the effect of microwave vacuum drying parameters on moisture migration, drying kinetics and quality characteristics of Angelica sinensis. In the research very comprehensive approach is presented, using equipment proposed and constructed by authors. All significant quality attributes are tested.
The manuscript topic is interesting and relevant for the journal scope, especially for its’ practical application.
The title of the manuscript is elaborative, informative and corresponds to the content of the work well.
Introduction section is appropriate and elaborate.
The Materials and Methods section needs some additional data.
The results and discussion section is mostly appropriate, with extensive correlation to other significant studies. Some additions are needed
Conclusion section is comprehensive and concludes and summarizes all the results presented in this research.
All corrections are noted in manuscripts’ pdf file.
Authors performed elaborate experimental plan that is suitable for further mathematical modeling. Hence, addition of developed mathematical models would greatly extend volume of this research paper, authors are suggested to develop mathematical models of the effects of drying temperature, vacuum pressure and sample thickness on tested responses and present them in successive study, since these models would have significant theoretical and practical value.
Reviewer recommendation: Major revision.

Reviewer 2 Report
I have included all comments for the authors in the attached file.

Reviewer 3 Report
The paper "Improving drying characteristics and physicochemical quality of Angelica sinensis by novel tray rotation microwave vacuum drying" corresponds to the chosen field. The paper is clear and well structured. To eliminate the shortcomings of uneven heating of traditional microwave drying and preserve the quality of Angelica sinensis, rotary microwave vacuum drying equipment was used. This paper describes the effect of temperature, slice thickness, and degree of vacuum on drying characteristics, physicochemical quality, and microstructure of dried Angelica sinensis продуктів. Загальний ефект фізико-хімічних властивостей після сушіння визначали за допомогою аналізу головних компонентів (PCA) і кореляційного аналізу Пірсона. Результати показали, що мікрохвильове вакуумне сушіння може значно скоротити час сушіння та підвищити ефективність сушіння для отримання високоякісних продуктів, і забезпечили теоретичну та наукову основу для вивчення та застосування в технології сушіння Angelica sinensis .
Є кілька коментарів до статті:
1. Необхідно описати приготування екстракту дягелю китайського .
2. Варто додати хроматограми ВЕРХ, які визначали вміст природних діючих речовин за різних умов попередньої обробки.
4. Необхідно розширити опис методики дослідження антиоксидантної активності (розділ 2.5.5).
5. Слід уточнити, що було використано як контроль для визначення антиоксидантної активності.
Рукопис актуальний і містить 54 посилання, 48 з яких належать до публікацій за останні 5 років, і лише 6 літературних даних старші 5 років.
Рукопис статті добре науково обґрунтований. Усі експериментальні результати показані як середнє ± стандартне відхилення. Висновки узгоджуються з отриманими результатами та аргументами.Заяви щодо етики та доступності даних є адекватними.
Reviewer 4 Report
The authors have tried to improve drying characteristics and physicochemical quality of Angelica sinensis by novel tray rotation microwave vacuum drying. The topic is original and relevant. The novelty of the paper is clearly indicated by addressing the challenges of uneven heating in microwave drying. The methodology section is ok. However, the conclusion need revision. Not all results be repeated in conclusion. Any limitation of the study to be mentioned in conclusion. The language needs minor improvement.
Round 2
Reviewer 1 Report
Authors have imporved manuscript quality. It is now suitable for publication.
Authos are also encouraged to reconsider insight noted in previous review:
Authors performed elaborate experimental plan that is suitable for further mathematical modeling. Hence, addition of developed mathematical models would greatly extend volume of this research paper, authors are suggested to develop mathematical models of the effects of drying temperature, vacuum pressure and sample thickness on tested responses and present them in successive study, since these models would have significant theoretical and practical value.
Reviewer 2 Report
The authors responded very thoroughly and correctly to my comments and questions.